# Prediction Models Using Decision Tree and Logistic Regression Method for Predicting Hospital Revisits in Peritoneal Dialysis Patients

**DOI:** 10.3390/diagnostics14060620

**Published:** 2024-03-14

**Authors:** Shih-Jiun Lin, Cheng-Chi Liu, David Ming Then Tsai, Ya-Hsueh Shih, Chun-Liang Lin, Yung-Chien Hsu

**Affiliations:** 1Department of Nephrology, Chang Gung Memorial Hospital, Chiayi Branch, Chiayi 613016, Taiwan; jiun0718@gmail.com (S.-J.L.); gnoo0102@gmail.com (C.-C.L.); d10705@hotmail.com (D.M.T.T.); linchunliang@cgmh.org.tw (C.-L.L.); 2Kidney and Diabetic Complications Research Team (KDCRT), Chang Gung Memorial Hospital, Chiayi 613016, Taiwan; 3School of Traditional Chinese Medicine, College of Medicine, Chang Gung University, Taoyuan 333423, Taiwan; 4Center for Shockwave Medicine and Tissue Engineering, Chang Gung Memorial Hospital, Kaohsiung 833253, Taiwan; 5College of Medicine, Chang Gung University, Taoyuan 333423, Taiwan

**Keywords:** peritoneal dialysis, decision tree, hospital revisits

## Abstract

Hospital revisits significantly contribute to financial burden. Therefore, developing strategies to reduce hospital revisits is crucial for alleviating the economic impacts. However, this critical issue among peritoneal dialysis (PD) patients has not been explored in previous research. This single-center retrospective study, conducted at Chang Gung Memorial Hospital, Chiayi branch, included 1373 PD patients who visited the emergency room (ER) between Jan 2002 and May 2018. The objective was to predict hospital revisits, categorized into 72-h ER revisits and 14-day readmissions. Of the 1373 patients, 880 patients visiting the ER without subsequent hospital admission were analyzed to predict 72-h ER revisits. The remaining 493 patients, who were admitted to the hospital, were studied to predict 14-day readmissions. Logistic regression and decision tree methods were employed as prediction models. For the 72-h ER revisit study, 880 PD patients had a revisit rate of 14%. Both logistic regression and decision tree models demonstrated a similar performance. Furthermore, the logistic regression model identified coronary heart disease as an important predictor. For 14-day readmissions, 493 PD patients had a readmission rate of 6.1%. The decision tree model outperformed the logistic model with an area under the curve value of 79.4%. Additionally, a high-risk group was identified with a 36.4% readmission rate, comprising individuals aged 41 to 47 years old with a low alanine transaminase level ≤15 units per liter. In conclusion, we present a study using regression and decision tree models to predict hospital revisits in PD patients, aiding physicians in clinical judgment and improving care.

## 1. Introduction

Hospital revisits result in increased medical costs and lead to the wastage of healthcare resources. To further explore this issue, hospital revisits can be categorized into readmissions and emergency room (ER) revisits. Research indicates that unscheduled ER revisits are associated with increased medical service use [1], suggesting that initial treatments failed to provide accurate diagnosis and effective care. ER revisits can contribute to crowding and lower the quality of care, making it a critical indicator for monitoring medical care quality [2]. Similarly, readmissions can lead to increased expenses for healthcare organizations and patients, as well as an elevated risk of severe complications and mortality. Therefore, readmission rates have become a primary performance indicator for assessing the quality of care provided to patients [3]. To address these issues, the Joint Commission of Taiwan (JCT) launched indicators, including 72-h ER revisits and the 14-day readmission rate, to assist hospitals in monitoring healthcare quality and to respond to external demands for accountability. Given the complex clinical characteristics and comorbidities among peritoneal dialysis (PD) patients, hospital revisit rates are expected to be higher than that of the general population. Therefore, developing strategies to decrease hospital revisits among PD patients could enhance the quality of care and reduce the financial burden. However, this critical issue in PD patients has not been addressed in any prior studies.

Prediction models in healthcare are a growing area of application of artificial intelligence (AI) in medicine [4]. Prediction models aim to develop algorithms and methods to learn from data and make predictions according to the data. Prediction applications are widely applied in healthcare to support doctors and independent decision-making mechanisms [4]. A variety of methods were developed to construct the prediction models. Among these methods, logistic regression and decision trees were the most commonly used [5,6]. Logistic regression is a statistical tool that fits the probability of an event by a linear function of the explanatory variables [5]. It is widely used because of its simplicity. On the other hand, the decision tree is a data analysis method known for its features such as classification and prediction [6]. The decision tree possesses advantages such as rapid establishment and easily understood if–then rules, making it commonly employed in various classification methods [6]. Despite previous studies on models used in predicting hospital revisits, none has focused on PD patients. In this study, we aim to use logistic regression and decision tree models to assist the clinical physicians in reducing the rate of hospital revisits among PD patients.

## 2. Materials and Methods

### 2.1. Data Source

We conducted this retrospective single-center study at the Chang Gung Memorial Hospital, Chiayi branch, Taiwan, utilizing electronic medical records (EMRs) from January 2002 to May 2018. The EMRs included patient demographics (age, sex), diagnosis, vital signs, lab data, medical cost, and relevant information in the ER (i.e., triage scale and Glasgow Coma Scale [GCS] score). The disease diagnoses were coded using the International Classification of Disease, Ninth Revision, Clinical Modification (ICD-9-CM) before 2016, and the International Classification of Disease, Tenth Revision (ICD-10-CM) thereafter. The Institutional Review Board of the Chang Gung Medical Foundation approved this study (Institutional Review Board Number: 201800837B0D001 and 202201910B0C501).

### 2.2. Study Design

The primary aim of this study was to explore the possible predictors of return to the ER within 72-h. Patients with PD who visited the ER and were not admitted between January 2002 and May 2018 were included in this study. The first ER episode of the patient during 2002 and 2018 was selected as the index ER if the patient had multiple ER episodes. As a consequence, a total of 880 PD patients were included, of whom 123 (14%) returned to the ER within 72-h after discharge of the index ER. The secondary aim was to explore the possible predictors of readmission within 14 days. Patients who were admitted for the index ER and survived the index hospitalization were included. A total of 493 patients were included, of whom 30 (6.1%) were admitted again within 14 days after discharge of the index admission.

### 2.3. Covariates and Outcomes

The covariates in this study were age, sex, Taiwan triage and acuity scale, and GCS upon arrival at the ER, 10 kinds of comorbidity (coronary heart disease [CHD], hypertension, diabetes mellitus [DM], stroke, chronic obstruction pulmonary disease, peripheral arterial disease, hyperkalemia, acute pulmonary edema, acidosis, and heart failure [HF]), vital signs (including blood pressure, body temperature, heart rate, and respiratory rate) upon arrival at the ER, and 13 sorts of lab data (leukocyte, hemoglobin, alanine aminotransferase [ALT], blood urea nitrogen, creatinine, sodium, potassium, c-reactive protein, calcium, phosphates, platelet, and neutrophil) upon arrival at the ER. Information of both the primary outcome (return to the ER within 72-h) and secondary outcome (readmission within 14 days) were available in the EMRs database.

### 2.4. Statistics

The baseline characteristics of patients with the presence or absence of outcomes (return to the ER within 72-h; readmission within 14 days) were compared using an independent sample t-test for continuous variables and chi-square test for categorical variables. To examine the association between the baseline characteristics and the risk of outcome, those variables whose significance was less than 0.2 in the previous univariate analyses were introduced into a further multivariable logistic regression model with a backward elimination to avoid overfitting. However, the multivariable logistic regression analysis is not advantageous in dealing with and presenting the possible interaction effects between the explanatory variables. Therefore, an alternative machine-learning model, namely the decision tree analysis, was adopted to present possible interactions among the predictors. Like the logistic model, only those variables whose significance was less than 0.2 were included in the decision tree analysis. There are several kinds of decision tree algorithms, such as the classification and regression tree (CART), chi-squared automatic interaction detection (CHAID), and quick unbiased efficient statistical tree algorithms (QUEST). Among these algorithms, CHAID was selected for its superior prediction accuracy [7]. We adopted an algorithm of exhaustive CHAID in which the maximum tree depth was 3 levels. Due to the limited event and sample size, the minimum cases in the parent and child nodes were set as 20 and 10, respectively. The results derived from the multivariable logistic regression analysis and decision tree analysis were compared by examining the difference in the area under the curves (AUCs). Two-tailed *p* values < 0.05 were considered statistically significant, and no adjustment of multiple testing (multiplicity) was made. All statistical analyses were performed using SPSS version 25.0 (IBM Corp., Armonk, NY, USA).

## 3. Results

### 3.1. Patient Characteristics for the ER Return Study

A total of 880 PD patients who visited the ER and were not admitted were analyzed in the study. Among these patients, 123 (14%) returned to the ER within 72-h and 757 (86%) did not. The mean age was 54.5 years (standard deviation [SD]: 13.2) years and approximately half of the cohort were male (54%). Only 42 patients (4.8%) were classified as resuscitation, 172 (19.5%) as emergency, and 527 (59.9%) as urgent. Almost all patients had hypertension (97.8%), followed by diabetes (45.6%), stroke and acute pulmonary edema (both 22.4%), heart failure (19.8%), and CHD (13.6%). The median duration of ER stay was 1.7 h (interquartile range: 0.9 to 5.2 h). Two patients (0.2%) died in the ER. Patients who returned to the ER within 72-h had a significantly higher GCS score, more prevalent CHD, and a short duration of ER stay compared to patients who did not (Table 1).

### 3.2. Prediction of ER Return within 72-h

The result of the multivariable logistic regression model showed that CHD was significantly associated with an increased risk of return to the ER within 72-h (odds ratio [OR]: 2.36, 95% confidence interval [CI]: 1.31–4.25) (Table 2).

In contrast, the result of the decision tree analysis identified a total of nine terminal nodes (TN). Patients were firstly classified into three subgroups by age. Patients with younger ages (<36.8 years) had a relatively low risk of ER return (8%; TN1). Patients of middle age (36.8 to 55.7 years) were further classified into two subgroups by CHD, in which those without CHD were further classified by the age of 48 years (TN2-3). Those middle-aged patients with CHD were further classified into three subgroups by the duration of ER stay, in which the patients whose duration was shorter (<1.4 h) or was longer (>2.3 h) had a relatively high risk of ER return (27.5% for TN4 and 32.3% for TN6). Patients with older ages (>55.6 years) were further classified into two subgroups by the duration of ER stay, in which the patients whose duration was longer (>1.1 h) had a low risk of ER return (8.1%; TN9). Those older-aged patients with a short duration in ER (<1.1 h) were further classified into two subgroups by sex, where the female patients had a high risk of ER return (25.5%; TN7) while the male patients had a moderate risk (11.1%; TN8) (Figure 1).

Figure 2 illustrates that the performance of discriminating the risk of return to the ER within 72-h was slightly better in the decision tree analysis (AUC: 65.6%, 95% CI: 60.4–70.7%) when compared to the logistic regression model (AUC: 61.5%, 95% CI: 56.4–66.6%).

Figure 3 displays the relative variable importance (VIMP) from the logistic regression and decision tree models. Using the most significant prediction variable as a reference, the findings indicate that CHD emerged as the most crucial predictor for the logistic regression model. Meanwhile, age and length of stay in the ER were identified as the two relatively important predictors for the decision tree model.

### 3.3. Patient Characteristics for the Readmission Study

A total of 493 PD patients who were admitted for the index ER and survived the index hospitalization were included, of whom 30 (6.1%) were admitted again within 14 days after discharge of the index admission and 463 (93.9%) were not.

The mean age was 59.0 years (standard deviation [SD]: 13.4) years and 47.1% were male. In total, 55 patients (10.6%) were classified as resuscitation, 149 (28.8%) as emergency, and 296 (57.1%) as urgent. Almost all patients had hypertension (97.2%), followed by diabetes (46.2%), hyperkalemia (29.0%), stroke (20.7%), heart failure (17.6%), and chronic obstruction pulmonary disease (15.0%). The median duration of the index admission stay was 6.9 days (interquartile range: 4.0 to 13.9 days). In total, 49 patients (9.9%) were transferred to the ICU and 28 (5.7%) removed the PD catheter during the index admission. Patients who were re-admitted within 14 days had significantly lower ALT and creatinine levels compared to patients who were not (Table 3).

### 3.4. Prediction of Readmission within 14-Days

The result of the multivariable logistic regression model showed that higher levels of ALT (OR: 0.96, 95% CI: 0.93–1.01, *p* = 0.083) and creatinine (OR: 0.89, 95% CI: 0.77–1.02, *p* = 0.099) were associated with borderline decreased risks of readmission within 14 days (Table 4).

The result of the decision tree analysis identified a total of six TNs. Patients were firstly classified into two subgroups by ALT level. Patients with lower ALT (≤15 U/L) were further classified into three subgroups by age (TN1-3), in which those of middle age (41.9 to 47.4 years) had a remarkably high risk of readmission within 14 days (36.4% for TN2). Patients with higher alanine aminotransferase (>15 U/L) were further classified into two subgroups by creatinine, in which those with higher creatinine (>8.8 mg/dL) had an extremely low risk of readmission within 14 days (0% for TN6). Patient with higher ALT (>15 U/L) and lower creatinine (≤8.8 mg/dL) were further classified by ALT level (TN4-5), where the patients with a higher ALT level had a relatively higher risk of readmission within 14 days (6.9% for TN5) and those whose ALT ranged from 15 to 20 U/L had an extremely low risk of readmission within 14 days (0% for TN4) (Figure 4).

Figure 5 illustrates that the performance of discriminating the risk of readmission within 14 days was substantially better in the decision tree analysis (AUC: 79.4%, 95% CI: 72.0–86.8%) when compared to the logistic regression model (AUC: 68.4%, 95% CI: 58.3–78.6%).

Figure 6 illustrates the relative VIMP from the logistic regression and decision tree models. Using the most significant prediction variable as a reference, the analysis reveals ALT and creatinine as the two relatively important predictors for both the logistic regression and decision tree model.

## 4. Discussion

Hospital revisits are known to cause increased medical costs and healthcare resource wasting [1]. Therefore, searches for strategies to lower the rate of hospital revisits are urgently needed. However, this critical issue has remained unexplored among peritoneal dialysis (PD) patients in previous research. In this study, we successfully utilized a decision tree and logistic regression model to predict hospital revisits for PD patients. We categorized hospital revisits into 72-h ER revisits and 14-day readmissions. For the 72-h ER revisits, both models had a similar prediction rate. Most importantly, the regression model identified CHD as an important predictor. For the 14-day readmission, the decision tree model outperformed the regression model with an AUC of 79.4%. We also identified a high-risk group with a readmission rate of 36.4%, which consisted of individuals with a low ALT level less than or equal to 15 (U/L) and of middle age (41 to 47 years old). In conclusion, this is the first study to demonstrate the effectiveness of prediction models in predicting hospital revisits among PD patients. The results of this study have significant implications for improving outcomes, optimizing the quality of care, and alleviating the financial burden of end-stage renal disease patients undergoing PD treatment.

Numerous studies in the past have developed prediction models for forecasting patient hospital revisits [8,9,10,11]. Different from previous research, our study incorporates a broader range of clinical parameters in our analysis, including laboratory tests, vital signs, GCS scores, and comorbidities as indicated in Table 1 and Table 3. This comprehensive analysis of patient clinical conditions has greatly increased the clinical relevance of our findings. We discovered CHD as the most important predictor for 72-h ER revisits, and ALT and creatinine levels as crucial predictors for 14-day readmissions, which could be utilized in clinical practice. By incorporating more clinical parameters, we gain a better understanding of the patients’ clinical conditions, giving our analysis more clinical applicability.

Our study finds that CHD is a reliable predictor for predicting ER revisits. In this study, after adjustment for confounding factor, multivariate logistic regression study showed that CHD is an independent risk factor of ER revisits. In addition, the relative VIMP analysis revealed CHD is the most important predictor for ER revisits. This result is consistent with a previous study by Masataka et al. [12] which showed that PD patients had a higher hospitalization rate due to CHD compared to HD patients. They found that PD is a risk factor for emergency hospitalization and mortality associated with CHD in patients with ESRD who undergo dialysis. The strict control of body fluid balance may help to prevent cardiovascular events in PD patients. In clinical implementation, this simple predictor helps physicians to quickly identify patients with a high risk of ER revisit and the priority of admission should be allocated accordingly, which is especially important in a crowded hospital with a shortage of medical resources. In the daily practice of ER physicians, it is important to conduct a detailed assessment of PD patients with a history of CHD before discharging them.

Our study indicates that the decision tree model is more effective than the regression model in predicting 14-day hospital readmission with an AUC of 79.4%. Additionally, we identified a subgroup with an extraordinarily high risk of readmission, characterized by low ALT levels less than or equal to 15 (U/L) and being of middle age (41 to 47 years old), with a remarkably high readmission rate of 36.4%. This decision tree model’s performance was good, as it identified this subset with a higher response rate than the rest of the population [13]. In clinical practice, it is important to conduct a more thorough evaluation of this high-risk population prior to discharge, given their increased risk of readmission. These findings underscore the significant benefits of using decision trees to manage the risk of hospital readmissions.

A notable and innovative finding in our prediction model for the 14-day readmission risk was that the two biomarkers, ALT and creatinine levels, were identified as important predictors in both logistic regression and the decision tree model, as revealed by the relative VIMP analysis. ALT is a transaminase enzyme that was formerly known as alanine aminotransferase or glutamate pyruvate transaminase [14]. It plays a crucial role in the alanine–glucose cycle by catalyzing the reversible transfer of an amino group from alanine to α-ketoglutarate, leading to the formation of pyruvate and glutamate. This process is essential for amino acid metabolism, gluconeogenesis, and cellular energy production [15]. Although ALT is commonly used to monitor liver tissue damage, its blood levels, as demonstrated by catalytic activity, have also been linked to whole-body skeletal muscle mass when the liver parenchyma is intact [16]. Previous research has demonstrated that lower ALT levels are a reliable marker of decreased muscle mass and frailty in large heterogeneous patient populations. Similarly, creatinine is a well-known marker of muscle mass [17]. Taken together, these findings suggest that insufficient muscle mass is a significant factor contributing to the risk of readmission. A rehabilitation program or nutrition intervention aimed at improving muscle mass could potentially lower the readmission rate.

An important question to consider is which prediction model is most suitable for predicting hospital revisits. For 72-h ER revisits, both the logistic regression model and decision tree yielded similar prediction rates. However, the regression model identified CHD as a significant predictor, making it more convenient to use in clinical practice. In contrast, regarding the 14-day readmission, the decision tree model demonstrated a superior performance compared to the logistic regression model. These results showed that each model is more suitable for a specific population.

With the advancement of AI and machine-learning (ML) technologies, many prediction models have made progress in predicting hospital revisit scenarios. LASSO (least absolute shrinkage and selection operator) is a significant improvement over traditional logistic regression models [18,19,20]. It is effective in situations where datasets are small compared to the number of variables, which can lead to overfitting. LASSO reduces regression coefficients and performs automatic variable selection by setting some coefficients to zero, thus alleviating overfitting [18]. Decision trees are favored for their simplicity and interpretability. However, as the volume of data increases, this simple model may face challenges. Thus, new models like random forests and eXtreme gradient boosting (XGBoost) have been developed. Random forests improve prediction accuracy by combining multiple randomized decision trees and averaging their predictions [21,22]. XGBoost is a boosting method that reduces prediction errors by sequentially adding models [10,23]. XGBoost builds upon the concept of decision trees by using them as the building blocks in a gradient boosting framework. Each tree in XGBoost tries to correct the errors of the previous trees, and the final prediction is made based on the ensemble of progressively refined trees. While centered on decision trees, XGBoost extends far beyond traditional decision tree models, offering solutions for a broad spectrum of regression, classification, and ranking challenges [23]. The voting classifier is an ensemble machine learning model that combines the predictions from different models to make a final prediction [8,24]. Deep-learning models, such as neural networks, aim to allow AI to autonomously discover data features and adjust models with large datasets [20,25]. Considering the relatively small number of patients, yet focusing on a specific patient group with more in-depth data (laboratory tests, vital signs, and GCS scores), we chose simple but interpretable models (decision tree and logistic regression models) to avoid overfitting and ensure clinical relevance. However, with AI and ML’s evolution, incorporating more patients and advanced models like random forests, XGBoost, voting classifiers, or deep-learning techniques could further improve the prediction accuracy in the future.

Some certain limitations should be acknowledged in this study. Firstly, our study is small-sample-size research which may not represent the general PD population. Further large cohorts like the Chang Gung Research Database (CGRD) or Taiwan National Health Insurance Research Database (NHIRD) are needed to investigate this issue. Secondly, for the small-sample-size population, we did not include a validation group. Therefore, further clinical use or external validation is needed to modify our prediction model. However, our data include complete demographic data, clinical parameters, and laboratory tests which makes our results reliable. Finally, the prediction performance of the AUC area is not sufficient which may relate to the small sample size. However, the importance of our study is that this is the first study to perform a prediction model for the prediction of PD patients which could be the pioneer in this area.

## 5. Conclusions

Hospital revisits are recognized as a significant concern in healthcare, yet this critical issue among PD patients has not been explored in previous research. In this study, we utilized logistic regression and decision tree models to predict hospital revisits, with a specific focus on PD patients. We also conducted an in-depth analysis of their condition during ER visits by incorporating data on vital signs, laboratory tests, and GCS scores. In the 72-h ER revisit analysis, both models had a comparable prediction rate. Additionally, we identified CHD as a significant predictor for 72-h ER revisits. For the 14 day-readmission analysis, the decision tree model outperformed the logistic regression model. Furthermore, we identified a group with a high readmission rate, consisting of patients aged 41 to 47 years with ALT ≤ 15 units per liter. This straightforward and interpretable rule could aid physicians in clinical judgment to decrease the hospital revisits of PD patients. However, due to the study’s single-center nature and relatively small sample size, its findings may not be universally applicable to all PD patients. Further large cohort studies such as CGRD or NHIRD research studies are necessary to develop a prediction model that can be more broadly applied in clinical practice.

## Figures and Tables

**Figure 1 diagnostics-14-00620-f001:**
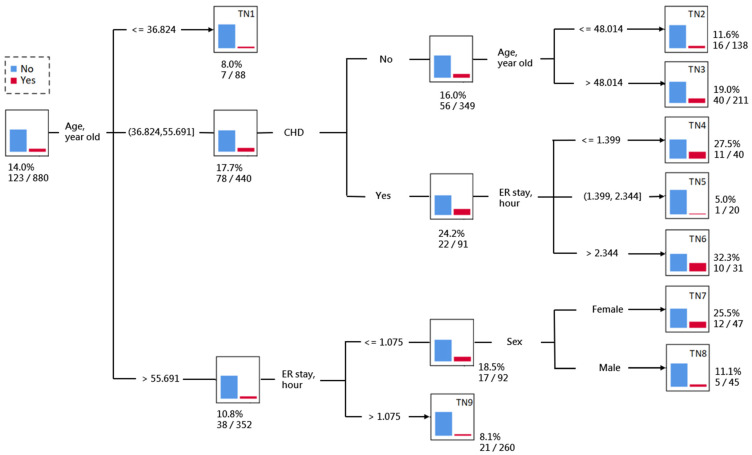
The decision tree of classifying the risk of return to the ER within 72-h. Abbreviations: CHD, coronary heart disease; ER stay, emergency room stay.

**Figure 2 diagnostics-14-00620-f002:**
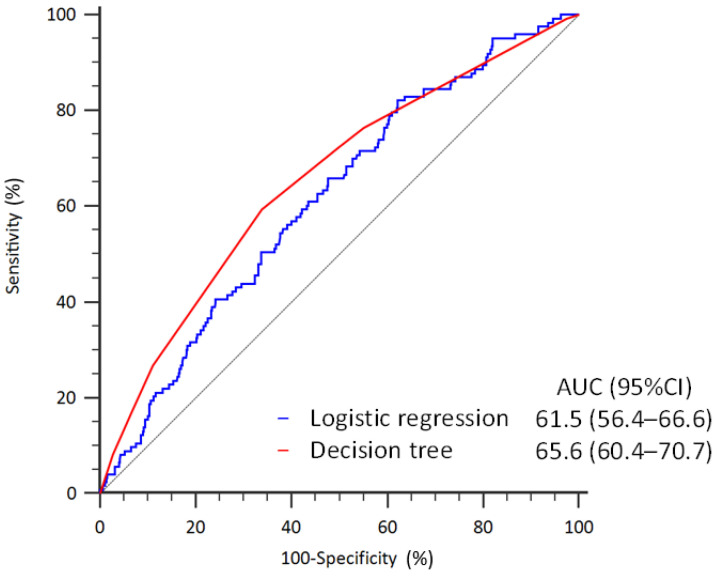
Receiver operating characteristic curves of discriminating the risk of return to the ER within 72-h in the logistic regression and decision tree models.

**Figure 3 diagnostics-14-00620-f003:**
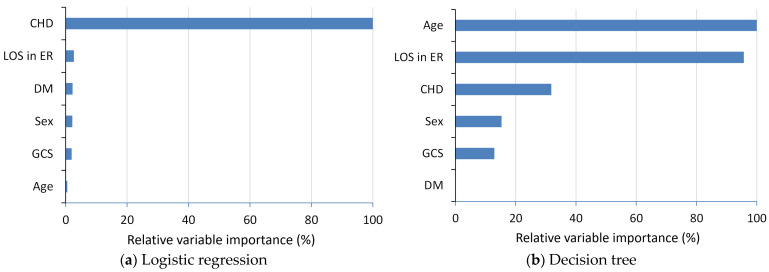
Relative variable importance (VIMP) from the (**a**) logistic regression model and (**b**) decision tree model in the ER return study. Abbreviations: CHD, coronary heart disease; LOS in ER, length of stay in emergency room; DM, diabetes mellitus; GCS, Glasgow Coma Scale.

**Figure 4 diagnostics-14-00620-f004:**
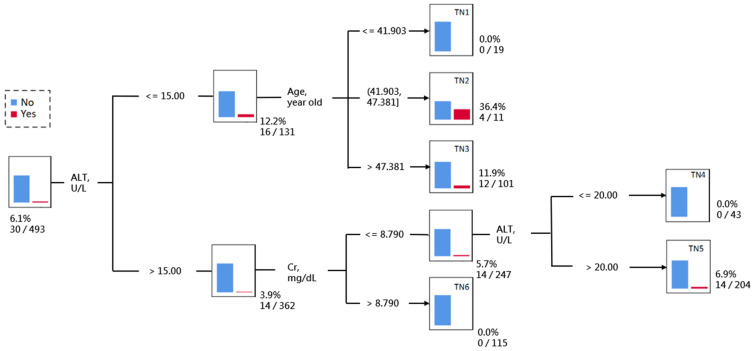
The decision tree of classifying the risk of readmission within 14 days. Abbreviations: ALT, alanine aminotransferase; Cr, creatinine.

**Figure 5 diagnostics-14-00620-f005:**
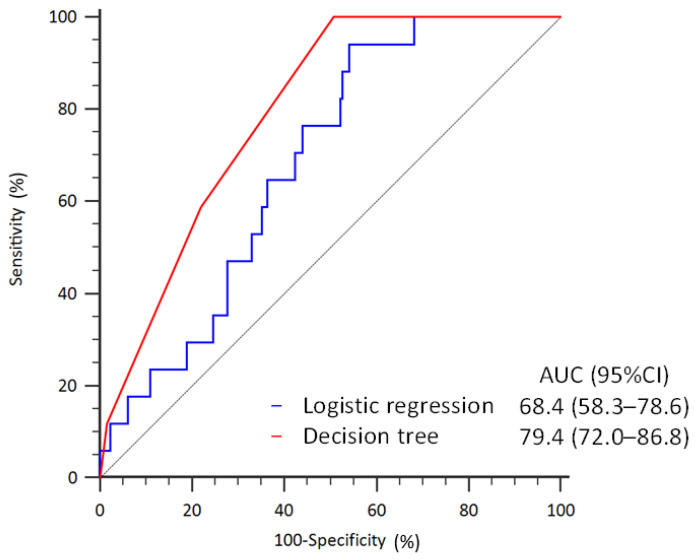
Receiver operating characteristic curves of discriminating the risk of readmission within 14 days in the logistic regression and decision tree models.

**Figure 6 diagnostics-14-00620-f006:**
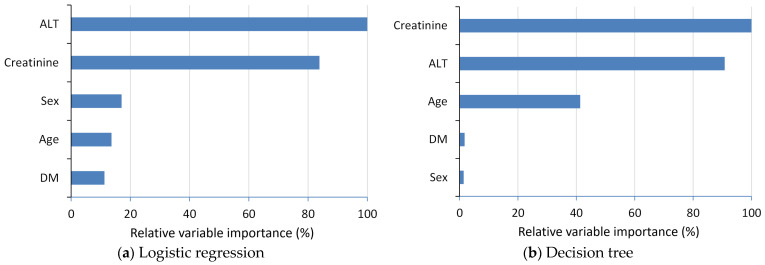
Relative VIMP from the (**a**) logistic regression model and (**b**) decision tree model in the 14-day readmission study. Abbreviations: ALT, alanine aminotransferase; DM, diabetes mellitus.

**Table 1 diagnostics-14-00620-t001:** Baseline characteristics of patients who were not admitted according to return to the ER within 72-h.

			Return to ER within 72-h		
Variable	Valid *N*	Total (*n* = 880)	Yes (*n* = 123)	No (*n* = 757)	*p* Value
Male	880	475 (54.0)	63 (51.2)	412 (54.4)	0.737
Age, year	880	54.5 ± 13.2	54.5 ± 11.8	54.5 ± 13.4	0.981
Taiwan triage and acuity scale	880				0.433
Level I: Resuscitation		42 (4.8)	6 (4.9)	36 (4.8)	
Level II: Emergency		172 (19.5)	26 (21.1)	146 (19.3)	
Level III: Urgency		527 (59.9)	74 (60.2)	453 (59.8)	
Level IV: Less urgency		122 (13.9)	17 (13.8)	105 (13.9)	
Level V: Non urgency		17 (1.9)	0 (0.0)	17 (2.2)	
Glasgow Coma Scale	878	14.8 ± 0.94	15.0 ± 0.09	14.8 ± 1.01	<0.001
Comorbidity					
Coronary heart disease	880	120 (13.6)	25 (20.3)	95 (12.5)	0.006
Hypertension	880	861 (97.8)	121 (98.4)	740 (97.8)	0.872
Diabetes mellitus	880	401 (45.6)	54 (43.9)	347 (45.8)	0.684
Stroke	880	197 (22.4)	30 (24.4)	167 (22.1)	0.813
Chronic obstruction pulmonary disease	880	105 (11.9)	11 (8.9)	94 (12.4)	0.287
Peripheral arterial disease	880	111 (12.6)	8 (6.5)	103 (13.6)	0.115
Hyperkalemia	880	267 (30.3)	38 (30.9)	229 (30.3)	0.690
Acute pulmonary edema	880	197 (22.4)	35 (28.5)	162 (21.4)	0.135
Acidosis	880	64 (7.3)	9 (7.3)	55 (7.3)	0.837
Heart failure	880	174 (19.8)	26 (21.1)	148 (19.6)	0.867
Vital signs					
Systolic blood pressure, mmHg	535	144.0 ± 35.0	140.5 ± 33.6	144.6 ± 35.2	0.300
Diastolic blood pressure, mmHg	534	83.0 ± 19.0	82.7 ± 16.5	83.3 ± 19.0	0.791
Body temperature, °C	330	36.5 ± 0.7	36.6 ± 0.7	36.4 ± 0.7	0.217
Heart rate, beat/min	529	82.2 ± 15.3	82.6 ± 14.2	82.1 ± 15.5	0.773
Respiratory rate, beat/min	163	21.6 ± 12.3	20.1 ± 0.4	21.9 ± 13.7	0.449
Peritonitis	880	38 (4.3)	7 (5.7)	31 (4.1)	0.423
Lab data					
Leukocyte, 1000/μL	522	7.3 (5.4, 9.7)	6.8 (4.9, 9.6)	7.3 (5.5, 9.7)	0.195
Hemoglobin, g/dL	534	10.0 (8.7, 11.3)	10.0 (8.5, 11.2)	9.9 (8.8, 11.3)	0.538
Alanine aminotransferase, U/L	409	20.0 (15.0, 28.0)	20.0 (16.0, 26.0)	20.0 (15.0, 28.0)	0.929
BUN, mg/dL	288	61.7 (42.2, 79.0)	62.6 (39.5, 77.8)	61.4 (42.5, 79.1)	0.735
Creatinine, mg/dL	293	10.1 (7.4, 12.9)	10.4 (8.8, 14.3)	10.0 (7.2, 12.9)	0.120
Sodium, mg/dL	493	134.6 (131.0, 137.2)	135.0 (130.8, 137.8)	134.3 (131.1, 137.1)	0.791
Potassium, mg/dL	506	3.8 (3.2, 4.3)	3.8 (3.1, 4.3)	3.8 (3.2, 4.3)	0.624
C-reactive protein, mg/dL	211	10.0 (5.0, 28.9)	12.8 (5.0, 45.2)	9.6 (5.0, 25.4)	0.076
Calcium, mg/dL	254	9.8 (9.2, 10.5)	9.9 (9.3, 10.6)	9.8 (9.1, 10.4)	0.191
Phosphates, mg/dL	201	4.8 (3.9, 5.7)	5.0 (4.2, 5.8)	4.8 (3.9, 5.7)	0.276
Platelet, 1000/μL	515	188.0 (151.0, 229.0)	189.5 (151.5, 229.5)	188.0 (149.0, 229.0)	0.750
Neutrophil, %	495	70.8 (62.0, 78.8)	71.0 (60.5, 79.0)	70.6 (63.0, 78.4)	0.754
Emergency room stay, h	880	1.7 (0.9, 5.2)	1.6 (0.6, 3.7)	1.8 (1.0, 5.4)	0.017
Death in emergency room	871	2 (0.2)	0 (0.0)	2 (0.3)	0.604

Abbreviations: ER, emergency room; BUN, blood urea nitrogen. Data are given as a frequency (percentage), mean ± standard deviation, or median (25th, 75th percentile).

**Table 2 diagnostics-14-00620-t002:** Multivariable logistic regression analysis for the associated factors of the risk of return to the ER within 72-h.

Predictor	OR (95% CI)	*p* Value
Male	0.75 (0.49–1.15)	0.185
Age, year	1.00 (0.99–1.02)	0.773
Coronary heart disease	2.36 (1.31–4.25)	0.004
Diabetes mellitus	0.73 (0.47–1.15)	0.177
Emergency room stay, hour	0.99 (0.97–1.01)	0.148
Glasgow Coma Scale	3.01 (0.55–16.46)	0.204

Abbreviations: ER, emergency room; OR, odds ratio; CI, confidence interval.

**Table 3 diagnostics-14-00620-t003:** Baseline characteristics of patients who were admitted for the index ER and survived the hospitalization according to readmission within 14 days.

			Readmission within 14 Days		
Variable	Valid *N*	Total (*n* = 493)	Yes (*n* = 30)	No (*n* = 463)	*p* Value
Male	493	232 (47.1)	15 (50.0)	217 (46.9)	0.739
Age, year	493	59.0 ± 13.4	57.8 ± 12.7	59.0 ± 13.4	0.613
Taiwan triage and acuity scale	493				0.771
Level I: Resuscitation		55 (10.6)	2 (6.7)	44 (9.5)	
Level II: Emergency		149 (28.8)	6 (20.0)	134 (28.9)	
Level III: Urgency		296 (57.1)	21 (70.0)	268 (57.9)	
Level IV: Less urgency		17 (3.3)	1 (3.3)	16 (3.5)	
Level V: Non urgency		1 (0.2)	0 (0.0)	1 (0.2)	
Glasgow Coma Scale	493	14.48 ± 1.66	14.37 ± 2.01	14.48 ± 1.64	0.709
Eye opening	488	3.95 ± 0.32	3.90 ± 0.55	3.95 ± 0.30	0.365
Verbal response	489	4.78 ± 0.73	4.63 ± 1.03	4.79 ± 0.70	0.411
Motor response	489	5.89 ± 0.47	5.73 ± 0.98	5.90 ± 0.41	0.369
Comorbidity					
Coronary heart disease	493	40 (8.1)	3 (10.0)	37 (8.0)	0.696
Hypertension	493	479 (97.2)	30 (100.0)	449 (97.0)	0.334
Diabetes mellitus	493	228 (46.2)	15 (50.0)	213 (46.0)	0.671
Stroke	493	102 (20.7)	5 (16.7)	97 (21.0)	0.575
Chronic obstruction pulmonary disease	493	74 (15.0)	5 (16.7)	69 (14.9)	0.793
Peripheral arterial disease	493	56 (11.4)	3 (10.0)	53 (11.4)	0.809
Hyperkalemia	493	143 (29.0)	9 (30.0)	134 (28.9)	0.901
Acute pulmonary edema	493	62 (12.6)	1 (3.3)	61 (13.2)	0.115
Acidosis	493	53 (10.8)	1 (3.3)	52 (11.2)	0.176
Heart failure	493	87 (17.6)	4 (13.3)	83 (17.9)	0.522
Vital signs					
Systolic blood pressure, mmHg	492	144.2 ± 34.9	146 ± 31	144.1 ± 35.2	0.740
Diastolic blood pressure, mmHg	492	80.5 ± 17.2	84.0 ± 13	80.3 ± 17.5	0.150
Body temperature, °C	493	36.8 ± 0.9	36.7 ± 0.8	36.8 ± 0.9	0.547
Heart rate, beat/min	492	86.9 ± 16.8	89.9 ± 15.4	86.7 ± 16.9	0.310
Respiratory rate, beat/min	373	21.4 ± 7.2	20.0 ± 1.0	21.4 ± 7.5	0.492
Peritonitis	493	159 (32.3)	13 (43.3)	146 (31.5)	0.180
Lab data					
Leukocyte, 1000/μL	485	9.6 (6.6, 13.0)	9.3 (7.0, 12.5)	9.6 (6.5, 13.0)	0.788
Hemoglobin, g/dL	483	9.6 (8.4, 10.8)	9.3 (8.3, 10.3)	9.6 (8.4, 10.8)	0.377
Alanine aminotransferase, U/L	415	20 (14, 32)	14 (8, 26)	21 (14, 32)	0.016
Albumin, g/dL	179	2.9 (2.3, 3.4)	2.6 (2.1, 3.1)	2.9 (2.4, 3.4)	0.161
BUN, mg/dL	306	57.2 (35.2, 77.3)	40.1 (30.5, 87.0)	57.9 (37.0, 77.3)	0.369
Creatinine, mg/dL	312	8.8 (6.4, 11.2)	7.0 (3.4, 9.5)	8.9 (6.5, 11.7)	0.025
Sodium, mg/dL	470	133 (130, 136)	134 (131, 136)	133 (130, 136)	0.465
Potassium, mg/dL	472	3.5 (2.9, 4.2)	3.8 (3.4, 4.4)	3.5 (2.9, 4.2)	0.061
C-reactive protein, mg/dL	233	40.7 (13.4, 96.2)	53.0 (34.3, 68.9)	39.3 (12.7, 100.8)	0.194
Calcium, mg/dL	266	9.3 (8.5, 10.0)	9.1 (8.3, 10.0)	9.3 (8.5, 10.0)	0.638
Phosphates, mg/dL	240	4.2 (3.1, 5.3)	3.3 (2.9, 4.6)	4.3 (3.1, 5.3)	0.310
Platelet, 1000/μL	481	200 (154, 253)	215 (149, 289)	200 (155, 252)	0.520
Neutrophil, %	480	78.7 (70.2, 85.0)	83.0 (75.7, 85.0)	78.5 (70.0, 85.0)	0.088
Admission days	493	6.9 (4.0, 13.9)	10.2 (6.7, 15.8)	6.8 (4.0, 13.9)	0.146
Transferred to ICU	493	49 (9.9)	3 (10.0)	46 (9.9)	0.991
Removal of PD catheter during hospitalization	493	28 (5.7)	3 (10.0)	25 (5.4)	0.291
Medical expenditure, ×10^3^ NTD	444	30.4 (15.3, 82.4)	59.4 (34.6, 118.8)	29.0 (15.0, 80.5)	0.828

Abbreviations: ER, emergency room; BUN, blood urea nitrogen; ICU, intensive care unit; PD, peritoneal dialysis; NTD, New Taiwan Dollar Data are given as a frequency (percentage) or the mean ± standard deviation.

**Table 4 diagnostics-14-00620-t004:** Multivariable logistic regression analysis for the associated factors of the risk of readmission within 14 days.

Predictor	OR (95% CI)	*p* Value
Male	0.67 (0.22–2.07)	0.487
Age, year	0.99 (0.95–1.03)	0.608
Diabetes mellitus	1.22 (0.39–3.81)	0.737
Alanine aminotransferase, U/L	0.96 (0.93–1.01)	0.083
Creatinine, mg/dL	0.89 (0.77–1.02)	0.099

Abbreviations: OR, odds ratio; CI, confidence interval.

## Data Availability

The authors declare that the data for this research are available from the correspondence authors upon reasonable request.

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
