# Peer review of "Prediction Models Using Decision Tree and Logistic Regression Method for Predicting Hospital Revisits in Peritoneal Dialysis Patients"

_diagnostics, 2024, doi:10.3390/diagnostics14060620_

Round 1
Reviewer 1 Report
Comments and Suggestions for Authors
It may be beneficial to include more detailed information about the AI model development process. Providing insights into the selection of algorithms and the reasoning behind their use could further enhance the quality of the paper.
The decision tree images in figures 1 and 3 are currently distorted and difficult to follow. I'm confident that restoring their quality would greatly improve their readability. Can you please look into this and see if it's possible to enhance the images? Also, include the feature impact of both algorithms to improve the readability of the discussion.
The references that were mentioned are a bit old. Can you please conduct a literature review on the most current papers available? The algorithms that were chosen, namely logistic regression and decision trees, have advanced features that enhance their predictive abilities. Could you also mention some of the latest approaches that have been developed? Additionally, I was curious as to why you did not cover any deep learning techniques in your review.
Reviewer 2 Report
Comments and Suggestions for Authors
This study investigated logistic regression and decision tree models for predicting hospital revisits in PD patients. ER visits of 1373 PD patients were analyzed and hospital readmissions were categorized into 72-hour ER visits and 14-day readmissions. For the 72-hour ED visit study, 880 PD patients had a visit rate of 14% and both logistic regression and decision tree models performed similarly. For 14-day readmissions, 493 PD patients were found to have a readmission rate of 6.1%. The decision tree model outperformed the logistic model with an area under the curve value of 79.4%. In this sense, the results of the study are satisfactory.
1- Figure 1 and 3 are not understandable at all, I wonder if there is a format shift.
2- What is the unit of the horizontal and vertical components of Figure 2?
3- The conclusion part of the study is incomplete (there is one missing)
4- A Discussion section should be added to the study.
5- The results of each table should be explained and discussed.
6- The motivation of the mastery of the study should be clearly emphasized in the conclusion.
Reviewer 3 Report
Comments and Suggestions for Authors
It is an interesting work, dealing with the important issue of Hospital revisits, but some improvements need to be made:
In the abstract: the presented subject is not clear. Additionally, sections labeled "Background:" or "Conclusions:" should be formatted into sentence structures to create a coherent ,that will be more readable.
The introduction section: requires a few adjustments in the structure of the information, and it would be better to include "Background:" at this point, as noted in the abstract. The dimensions in the images should be adjusted according to the specifications of [specific reference].
In the "2.1. Data source" section: does not clearly stated that the data were obtained from your own studies, and it should be clarified how much time was spent overall and whether all procedures were conducted with patient/hospital consent.
The discussion: is extensive, while the conclusion is overly brief. Information about how you worked on the paper could be moved to the conclusion from the discussion.
In the conclusion section: are not mentioned any ideas for future research and development on the topic, or at least reference if such ideas have already been discussed and are under consideration.
Round 2
Reviewer 2 Report
Comments and Suggestions for Authors
I believe that this study has become quite adequate after the implementation of the referee's suggestions. In this sense, I kindly suggest that the paper be accepted.